# DNA Damage Response in Multiple Myeloma: The Role of the Tumor Microenvironment

**DOI:** 10.3390/cancers13030504

**Published:** 2021-01-28

**Authors:** Takayuki Saitoh, Tsukasa Oda

**Affiliations:** 1Department of Laboratory Sciences, Graduate School of Health Sciences, Gunma University, 3-39-22 Showa-machi, Maebashi, Gunma 371-8511, Japan; 2Laboratory of Molecular Genetics, Institute for Molecular and Cellular Regulation, Gunma University, 3-39-15 Showa-machi, Maebashi, Gunma 371-8512, Japan; toda@gunma-u.ac.jp

**Keywords:** multiple myeloma, DNA repair, genomic instability, DNA damage response, base excision repair, homologous recombination

## Abstract

**Simple Summary:**

Multiple myeloma (MM) is a plasma cell malignancy. Novel therapies have improved outcomes in MM patients; however, MM still remains incurable. MM cells present genomic instability, whose molecular basis is not fully understood. Recently, it has been reported that the DNA damage response (DDR) may influence genomic changes and drug resistance in MM. An abnormal DNA repair function may provide an alternative explanation for genomic instability. In this review, we show an overview of the DNA repair pathways and discuss the role of the tumor microenvironment in governing the DNA repair mechanisms. The tumor microenvironment factors, such as hypoxia and inflammation, and chemotherapeutic agents profoundly influence the DNA repair pathways in MM.

**Abstract:**

Multiple myeloma (MM) is an incurable plasma cell malignancy characterized by genomic instability. MM cells present various forms of genetic instability, including chromosomal instability, microsatellite instability, and base-pair alterations, as well as changes in chromosome number. The tumor microenvironment and an abnormal DNA repair function affect genetic instability in this disease. In addition, states of the tumor microenvironment itself, such as inflammation and hypoxia, influence the DNA damage response, which includes DNA repair mechanisms, cell cycle checkpoints, and apoptotic pathways. Unrepaired DNA damage in tumor cells has been shown to exacerbate genomic instability and aberrant features that enable MM progression and drug resistance. This review provides an overview of the DNA repair pathways, with a special focus on their function in MM, and discusses the role of the tumor microenvironment in governing DNA repair mechanisms.

## 1. Introduction

Multiple myeloma (MM), which accounts for 1% of all cancers and approximately 10% of all hematologic malignancies, is characterized by the clonal proliferation of plasma cells [1,2]. The development and introduction of therapeutics, such as the immunomodulatory drugs thalidomide and lenalidomide and the proteasome inhibitor (PI) bortezomib, have led to improved prognosis of MM [1]. Before the introduction of alkylating agents in MM treatment, the median survival time of symptomatic patients was <1 year, and the introduction of melphalan in the 1960s resulted in improved survival [3,4]. High-dose melphalan (HDM) followed by autologous stem cell transplantation (ASCT) has become a standard of care for younger patients after bortezomib-based induction regimens [5,6]. In addition, the introduction of novel agents, particularly bortezomib combined with lenalidomide plus dexamethasone, has improved the outcome of patients who are ineligible for ASCT [7]. Recently, carfilzomib [8], pomalidomide [9], panobinostat [10], ixazomib [11], elotuzumab [12], daratumumab [13], isatuximab [14], and selinexor [15] have been approved by the Food and Drug Administration (FDA) for the treatment of relapsed MM and promise to improve outcomes. Currently, numerous combination therapies are available and include immunomodulatory drugs, PIs, histone deacetylase inhibitors, and monoclonal antibodies. However, MM remains an incurable disease, and new therapeutic strategies are still needed [1].

MM cells present genomic instability [16], whose molecular basis is not fully understood. Recently, it has been reported that the DNA damage response (DDR) may influence genomic changes in MM [17,18,19,20]. An abnormal DNA repair function may provide an alternative explanation for aneuploidy and chromosomal rearrangements. Furthermore, the tumor microenvironment may be mutagenic and constitute a significant source of genetic instability, affecting genomic stability and tumor resistance to treatment (Figure 1) [21,22,23]. This review provides an overview of the DDR, with a special focus on its function on MM. We will also discuss the role of DNA repair in regulating the metabolism and progression of MM cells.

## 2. Genomic Damage

Genomic stability is important for ensuring the inheritance of correct genetic information. The occurrence of genomic instability can result in harmful consequences, which might lead to severe diseases, such as cancers. Endogenous and exogenous environmental factors can act as DNA-damaging agents, causing genetic mutations [23,24]. Endogenous factors include natural metabolic byproducts, such as those resulting from oxidation, alkylation, hydrolysis, and mismatch of DNA bases, while exogenous factors include ionizing radiation, ultraviolet (UV) radiation, and various chemical agents [24,25]. These factors have been associated with the etiology of human cancer [23,24,26], with the induced DNA damage being an important first step in the process of carcinogenesis. Noteworthy, each human cell is estimated to endure approximately 70,000 DNA damage lesions per day [24]. Damaged DNA that is not properly repaired can then lead to genomic instability, apoptosis, or senescence [27]. Therefore, DNA repair systems are important to protect DNA integrity, as the resolution of DNA damage lesions is critical for normal cells. DNA repair systems involve cell cycle checkpoints, removal of mutagenic lesions from DNA, and apoptosis or senescence if DNA repair fails [23,24], avoiding error information and interrupting neoplastic transformation [27].

## 3. Genomic Instability in MM

Most cancers are characterized by genomic instability [27,28,29], as cells often display chromosomal translocation, aneuploidy, and unchecked cell proliferation. Genomic instability is also a hallmark of MM cells, manifesting largely as whole chromosome- or translocation-based aneuploidy [22,30]. MM cells present various forms of genomic instability, including chromosomal instability (CIN), microsatellite instability, and increased frequency of mutations [31]. Especially, copy number and structural changes due to CIN are common features of MM. Numerical CIN is associated with copy number alterations (CNAs) of entire chromosomes or amplifications and deletions of chromosome arms [32]. The MM genome is also characterized by aberrations caused by structural CIN, such as chromosomal rearrangements, inversions, or complex reassembly of different parts of the chromosomes [32]. Centrosomes play important roles in processes that ensure proper segregation of chromosomes in human cells [33]. Chromosome abnormalities may be associated with centrosome amplification or alterations in the spindle assembly checkpoint. A high centrosome index—prompted by an abnormally high expression of genes encoding the main centrosome proteins—showed an independent prognostic factor in MM [34]. Moreover, dysregulations of the cell-cycle also affect CIN, the most prominent example of which is dysregulation of cyclin D expression—present in both hyperdiploid and non-hyperdiploid MM [31,35]. Furthermore, CIN is affected by the tumor microenvironment, such as hypoxia [36]. Depending on the genes affected, CIN can contribute to a further increase in genome instability [31].

MM patients are broadly grouped into hyperdiploid or non-hyperdiploid groups, depending on the number of chromosomes present in the MM cells. Hyperdiploid tumors are characterized by trisomies of one or more of the odd-numbered chromosomes 3, 7, 9, 11, 15, or 17, while the majority of non-hyperdiploid tumors display a translocation involving the *IGH* locus on chromosome 14 and one of the five recurrent translocation partners on chromosomes 4, 6, 11, 16, and 20. Five chromosomal partners account for the majority of primary *IGH* translocations: 11q13 (*CCND1*), 6p21 (*CCND3*), 4p16 (*FGFR3* and *NSD2*), and 16q23 (*MAF*) [37,38]. These events are present in most patients with monoclonal gammopathy of undetermined significance (MGUS), and secondary genetic alterations occur with an increased incidence in disease progression from MGUS to MM [39]. These secondary events include translocations, deletions, and chromosome gains, involving genes such as *MYC*, *KRAS*, *NRAS*, and *TP53*, which are involved in the DDR. Secondary events are detected in the late stage of the disease and affect its progression [39,40]. Among these alterations, t (4:14) and del(17p) were associated with poor outcomes [41]. The complexity of the genomic alteration characteristics is correlated with different grades of CIN. These observations strongly implicate CIN as an important biological and prognostic marker in MM [31,42].

Current treatment modalities of MM include alkylating agents, PIs, and anthracyclines, all of which induce excessive DNA damage [23]. DNA-damaging agents, including chemotherapy, can induce mutations and destabilize the genome, resulting in secondary malignancies or allowing the selection of resistant cancer cell clones [23,24]. Melphalan is a nitrogen mustard known to induce mono-alkylation of adenine and guanine, and interstrand DNA crosslinks (ICLs) involving guanine. The number of melphalan-induced ICLs is correlated with its concentration [43]. Cyclophosphamide is also an alkylating agent that induces ICLs. In a cohort of patients with MM treated with HDM and ASCT, polymorphisms in DNA repair genes, including poly(ADP-ribose) polymerase1 (*PARP1*), *RAD51*, *PCNA*, *OGG1*, *XPC*, *BRCA1*, *ERCC1*, *BARD1*, and *TP53BP1*, were associated with the outcome and overall survival [44]. These genes are significantly enriched in genes involved in homologous recombination repair (HRR) and nucleotide excision repair (NER), necessary for ICL repair. Bortezomib is the first PI approved by the FDA for the treatment of newly diagnosed and relapsed/refractory MM. This drug can bind to and form a complex with the active site of the threonine hydroxyl group in the β5 subunit of the proteasome and block its chymotrypsin-like activity [45]. Recent studies on proteasome inhibition in MM cells have revealed that the accumulation of unfolded proteins in the endoplasmic reticulum, the so-called endoplasmic reticulum stress, triggers that of several pro-apoptotic factors and cell stressors, such as reactive oxygen species (ROS). PI affects DNA repair via depletion of the free ubiquitin pool that is critical for further protein ubiquitination for building DNA repair foci through protein recruitment and degradation. Bortezomib treatment may prevent DNA resection by inhibiting the proteasomal degradation of proteins involved in chromatin relaxation, thus preventing the recruitment of replication protein A (RPA) onto single-stranded DNA (ssDNA) [46,47,48]. In vitro studies showed that bortezomib can possess synergistic anti-myeloma effects with melphalan by sensitizing MM cells to chemotherapeutic agents [49,50]. In clinical studies, bortezomib in combination with melphalan is efficacious for MM, both in the untreated and relapsed settings [51,52]. In the ASCT setting, the Intergroupe Francophone Du Myeloma (IFM) Phase II study showed significantly better outcomes in the ASCT recipients who received bortezomib–HDM conditioning compared with those treated with HDM only [53]. However, a recent analysis of the IFM 2014-02 trial Phase III study, in which 300 randomized patients received upfront ASCT to bortezomib and HDM, showed no difference in CR rate, PFS, and OS [54]. Doxorubicin is one of the most effective chemotherapy drugs for the treatment of many cancers, including lung cancers, leukemia, lymphoma, and MM [55]. This chemotherapeutic exerts its effects on cancer cells by intercalating into DNA, which results in DNA synthesis inhibition. Moreover, doxorubicin is a DNA topoisomerase II inhibitor, leading to DNA strand breaks by forming a cleavable complex with DNA and DNA topoisomerase II [56] and ROS formation in cells. Alkylating agents and DNA topoisomerase II inhibitors pose the risk of secondary cancers, such as therapy-related acute myeloid leukemia. Compared with de novo acute myeloid leukemia, the therapy-related disease is associated with poor prognosis and abnormal karyotypes, including the −5, −7, abnl(17p), complex karyotypes, and monosomal karyotypes [57].

DNA damage requires effective DNA repair capacity, which may be limited in tumor cells. Compared to normal cells, cancer cells present a higher accumulation of DNA damage and replication stress (RS) due to faulty cell cycle checkpoint activation [23,25]. Sources of RS include fragile sites, replication-transcription complex collision, secondary DNA structures, depletion of replication factors and nucleotides, and oncogenic stress [58]. Furthermore, DNA repair activity is required to counteract oxidative DNA damage in tumor cells induced by the tumor microenvironment. Cancer cells utilize mutagenic repair pathways for their advantage and to escape death [23]. Such repair pathways in cancer cells may represent treatment targets for their sensitization to drugs. Indeed, DNA repair inhibition has been used as a strategy for cancer treatment. Therapeutic strategies that target DNA damage include the use of PARP inhibitors for *BRCA1*/*BRCA2*-mutated cancers [59]. Thus, DNA repair pathways, including HRR, may be the target of drugs used to treat MM and play a role in resistance [60,61]. Next, we provide an overview of the DDR and major DNA repair pathways.

## 4. DDR

The DDR refers to a network of intracellular pathways that sense and resolve damaged DNA, but can also be a major source of genomic instability, particularly when cell death pathways are deactivated. This mechanism involves DNA repair, activation of cell cycle checkpoints, transcriptional modulation, apoptotic pathways, senescence, and DNA damage tolerance (Figure 2) [62,63,64,65,66,67]. The DDR signaling pathways consist of (1) signal sensors, which are proteins that recognize the DNA structure induced by DNA damage and RS; (2) transducers, which are kinases, including ataxia-telangiectasia mutated (ATM) and ATM/RAD3-related (ATR), and their downstream kinases; and (3) effectors, which are substrates of ATM, ATR, and their downstream kinases [63,68].

Molecular “sensors” of DNA damage lesions, such as DSBs, are early players in the DDR. The MRE11/RAD50/NBN (MRN) complex is a DSB sensor in the ATM pathway. This complex recruits ATM to DSB sites, and the interaction with NBN activates ATM, resulting in the phosphorylation of target proteins, such as the histone variant H2AX, which leads to γ-H2AX. Ku (comprising the KU70/KU80 protein heterodimer) is another DNA damage sensor that binds to DSBs and recruits DNA-PKcs. Autophosphorylation of DNA-PKcs serves as a platform for the subsequent recruitment of classical non-homologous end joining (NHEJ) proteins. RPA, a sensor in the ATR pathway, binds to ssDNA generated at the site of stalled replication forks. In turn, ATR-interacting protein (ATRIP) binds to the RPA–ssDNA complex to recruit ATR. The RAD9A/RAD1/HUS1 (9-1-1) complex is another DNA damage sensor in the ATR pathway. Regarding “transducers”, ATM/ATR and the downstream CHEK1/CHEK2 kinases are well-studied players in the DDR pathway. The DDR is controlled by ATM and ATR, both of which are activated by DNA damage and RS, but their functions differ. ATM is mainly implicated in the repair of DSBs, whereas ATR responds to various types of DNA damage that have in common the presence of ssDNA [64,69]. ATM and ATR then activate downstream kinases, such as the checkpoint kinases CHEK1 and CHEK2, through phosphorylation. CHEK1 is mainly activated through ATR-mediated phosphorylation and, in turn, phosphorylates CDC25A, leading to ubiquitination and proteasome-dependent protein degradation, and increased phosphorylation of cyclin-dependent kinase 2 (CDK2) downstream. CHEK2 is mainly activated by ATM and also phosphorylates CDC25A, leading to its rapid degradation and cell cycle arrest. Activated CHEK1 and CHEK2 phosphorylate diverse downstream effectors, which in turn are involved in cell cycle checkpoints (G1/S-phase, intra-S-phase, and G2/M-phase checkpoints), DNA replication checkpoints, mitotic checkpoints, DNA repair, and apoptosis [64,67]. ATM/ATR signaling also enhances DNA repair by inducing DNA repair proteins transcriptionally or post-transcriptionally via modulation of their phosphorylation, acetylation, ubiquitylation, or sumoylation [64,70].

MM cells present constitutive, ongoing DNA damage, as evidenced by the high number of γ-H2AX foci in their nuclei [18]. Furthermore, the oncogene MYC induces DNA damage in MM cells through increasing not only RS but also oxidative stress, triggering further DNA damage and apoptosis [20]. Recently, ATR inhibition with the compound VX-970 showed a highly synergistic effect with ICL-inducing melphalan in resistant MM cell lines. This combination was dramatically effective in targeting primary patient-derived myeloma cells and tumors in mouse models [71]. The checkpoint kinase inhibitor AZD7762 showed chemotherapy-induced apoptosis of p53-mutated multiple myeloma cells [72]. In addition, high levels of DNA damage were shown to make tumor cells dependent on a proper DDR [23,61,64,73]. This suggests that homologous recombination (HR) in the DDR could be a promising target for the treatment of MM.

## 5. DNA Repair Pathways

Mammalian cells have six major DNA repair pathways involved in the DDR: the base excision repair (BER), NER, and mismatch repair (MMR) pathways repair nucleotide lesions on ssDNA; the HR and NHEJ pathways are involved in DSB repair; and the Fanconi anemia (FA) pathway repairs ICL lesions in co-operation with the NER and HR pathways [23,24,25].

## 6. Major Single-Strand Break (SSB) Repair Pathways

Macrophages in the tumor microenvironment release proinflammatory cytokines and lead to genomic instability through the production of ROS and reactive nitrogen species. Such chemicals can attack DNA, leading to adducts that impair base pairing and/or block DNA replication, base loss, or DNA SSBs [64]. The latter is efficiently repaired by specific SSB repair pathways [73].

### 6.1. BER Pathway

SSBs occur more than 10,000 times per mammalian cell each day, being the most common type of DNA damage in cells [73]. These lesions can arise from direct attack by intracellular metabolites, oxidized bases during oxidative stress, intermediate products of BER pathways, and aborted activity of DNA topoisomerase I [74]. Unrepaired SSBs are located in the nucleus and mitochondria and may result in DNA RS, transcriptional stalling, and excessive PARP1 activation, leading to genomic instability. PARP is triggered by SSBs and DSBs, which recruit the DNA repair machinery through the synthesis of poly(ADP-ribose) chains [75]. BER recognizes DNA bases damaged by oxidation, deamination, and alkylation. Moreover, ROS induce the production of various types of oxidative DNA damage, some of which are implicated in mutagenesis. Among the various types of oxidative DNA damage, 8-hydroxyguanine is a highly mutagenic DNA lesion yielding G:C → T:A transversion, since 8-hydroxyguanine allows the incorporation of adenine as well as cytosine in DNA at the opposite site of the lesion. DNA glycosylase initiates BER by recognizing and removing damaged bases, which are processed by the APEX1endonuclease and later restored by DNA polymerase and a ligase [76,77,78].

APEX1 is a multifunctional protein involved both in the BER of DNA lesions and regulation of gene expression as a redox co-activator of different transcription factors, such as EGR1, TP53, NFKB1, HIF1A, PAX5, PAX8, and AP-1 [79,80]. APEX1is induced by oxidative agents, such as H_2_O_2_ and HOCl, and ROS-generated injuries, such as those caused by UV radiation. Increased expression of APEX1 correlates with an increase in its endonuclease and redox activities [76,79]. APEX1 is overexpressed in many cancer cells, including cervical [81], ovarian [82], and prostate [83] cancer cells and myeloma cells [84,85], and its altered level or intracellular distribution has been found in various cancers. Furthermore, high levels of APEX1 expression are associated with chemotherapy and radiotherapy resistance and poor prognosis [80,86,87]. Melphalan-resistant MM cell lines also show high expression of APEX1, and knocking down *APEX1* sensitizes them to melphalan treatment [84]; also, downregulation of *APEX1* in RKO cell lines using shRNA reduces cell proliferation. In addition, inhibition of APEX1 in CHO cells under acidic conditions increases oxidative DNA damage related to increased intracellular ROS [88]. Inhibition of the *APEX1* gene or protein by shRNA or an APEX inhibitor, respectively, reduces the proliferation of the MM cells. Noteworthy, APEX1 contributes to the dysregulation of other DNA repair pathways, such as HRR in MM cells [85]. Polymorphisms in the *APEX1* gene are also associated with shorter survival of patients with MM [89]. In addition to APEX1, APEX2 plays essential roles in the BER and ATR–CHK1 DDR pathways [90]. *APEX2* genomic alterations are estimated to occur with a frequency of ~17% in 14 cancer types, based on data available at The Cancer Genome Atlas. APEX2 expression is upregulated in several cancers, including kidney, breast, lung, liver, and uterine cancers. Furthermore, *APEX2* mRNA levels positively correlated with PCNA, APEX1, XRCC1, PARP1, CHEK1, and CHEK2 across these tumor tissues. Both APEX1 and APEX2 are upregulated in MM cell lines and samples from patients with MM [81], and in MM compared to MGUS [85,90]. shRNA-mediated knockdown of both *APEX1* and *APEX2* also inhibits HR activity. Based on this, upregulation of BER in cancers may represent an adaptive survival response [82,83].

PARP1 is a nuclear DNA-dependent ADP-ribosyltransferase that catalyzes the poly ADP-ribosylation (PAR) of nuclear proteins. PARP enzymes are broadly involved in the cellular response to DNA damage [91]. PARP1 has been shown to contribute to the progression of some cancers due to its fundamental role in cellular events, such as DNA damage repair, transcription, cell cycle progression, unfolded protein response, and cell death [91,92,93]. PARP1 regulates NSD2 via PARylation upon oxidative stress and affects the DSB repair pathways repair pathway [94]. NSD2, a histone methyltransferase, overexpressed in t (4;14)+ MM cells, leads to increased cell proliferation and altered downstream gene expression due to a genome-wide increase in H3K36 dimethylation and a decrease in gene repression-associated H3K27 trimethylation(H3K27me3).

High expression of PARP1 is associated with a higher grade of some cancers, including gastric, ovarian, and breast cancers [95,96,97]. The PARP1 V762A polymorphism reduces the enzymatic activity of PARP1 and increases the risk of many cancers, such as gastric, cervical, and lung cancers [98,99]. PARP inhibitors have been used either as single agents or in combination with DNA-damaging agents [100,101,102]. As stand-alone agents, PARP inhibitors have been applied to induce synthetic lethality in HR-deficient tumors. In addition, PARP inhibitors are effective in *BRCA1*- or *BRCA2*-deficient breast and ovarian cancers [102,103]. In MM, high PARP1 expression is correlated with poor survival, based on data of newly diagnosed patients with MM from the Arkansas dataset [104]. Olaparib, a PARP inhibitor, significantly reduces the proliferation of MM cell lines. Moreover, *PARP1* knockdown or olaparib was shown to result in significant inhibition of tumor growth using xenografts of human MM cells [105].

### 6.2. NER Pathway

The NER pathway removes the bulky DNA lesions induced by UV radiation, smoking, environmental mutagens, and alkylating agents [106]. This DNA repair mechanism consists of two pathways: global genomic NER (GG-NER) removes DNA damage from the whole genome, and transcription-coupled NER (TC-NER) specifically repairs transcription-blocking lesions in actively transcribed DNA [107]. After the DNA damage recognition step, GG-NER or TC-NER converge onto the same path to excise the damaged fragment and synthesize a new DNA strand portion. GG-NER is initiated by the recognition of damage-induced DNA helix distortions, while TC-NER is initiated by the stalling of RNA polymerase II at the lesion [107]. The NER gene expression profiles between normal plasma cells and samples of newly diagnosed MM show that high ERCC3 expression is associated with poor survival. Furthermore, 34 NER-related genes were found to be differentially expressed between MM and normal plasma cells, and 23 genes affected by copy-number alterations. Of note, removal of adducts generated by alkylating agents requires NER. MM cell lines with high rates of NER tend to be resistant to melphalan. Accordingly, NER inhibition significantly increases the sensitivity and overcomes resistance to alkylating agents in MM [108]. Moreover, ERCC1 and ERCC2 are involved in NER. The polymorphisms ERCC2 K751Q and XRCC3 T241M were shown to affect treatment outcomes in patients with MM treated with a high dose of ASCT [109].

### 6.3. MMR Pathway

The MMR pathway corrects replication errors, such as mismatched bases and insertion/deletion loops. MMR recognizes and repairs erroneous insertion, deletion, and misincorporation of bases that can arise during DNA replication, recombination, and repair of DNA damage. This pathway recruits DNA repair proteins to damaged sites and increases the accuracy of the DNA replication [110]. MMR promotes a DDR mediated by ATM and ATR in response to various types of DNA damage [110]. MMR is also implicated in the repair and cytotoxicity of a subset of DNA lesions caused by DNA alkylators, 6-thioguanine, cisplatin, and UV light [111,112,113]. Deficiency in MMR proteins results in a higher mutation rate, particularly in microsatellite DNA regions. Microsatellite instability, a manifestation of defective MMR, is found in many patients with MM [114]. In addition, MMR defects are found in MGUS/SMM/MM and increase their frequency during more active stages of the diseases. A microsatellite analysis showed instability at one or more of nine loci found in 15 of 92 patients: 7.7% of MGUS/SMM, 20.7% of MM/plasma cell leukemia, and 12.5% of relapsed MM/plasma cell leukemia [115].

## 7. DSB Repair Pathways

DSBs are created by genotoxic agents commonly used in the treatment of cancer. Radiation, oxidative free radicals, topoisomerase inhibitors, and DNA replication inhibitors can all lead to DSBs [23,116]. Failure to properly respond to genotoxic stress can be mutagenic and fatal to cells. DSBs caused by these agents can be resolved through the DSB repair pathways described in the next subsections.

### 7.1. HRR Pathway

HR is widely used to ensure genomic stability when a homologous template is available for repair. The MRN complex plays an important role in this DDR pathway, regulating both signaling and damage responses. MRN recognizes DSBs and activates ATM and ATR signaling, which orchestrate cell cycle progression and damage responses [117,118,119]. DNA end resection is coordinated by MRN proteins, resulting in single-stranded 3′ overhangs. The exposed ssDNA is coated with RPA and activates the ATR response to facilitate HRR. Recombination is initiated when the RAD51 recombinase actively displaces RPA to form the presynaptic filament that searches for and invades the homologous template. End processing is completed by excision repair cross-complementing group 1, and DNA end gaps are filled by DNA polymerase [24,120,121].

### 7.2. NHEJ Pathway

NHEJ is another major pathway that repairs DSBs in mammalian cells. NHEJ mediates the direct re-ligation of the broken DNA molecule [23,122]. This pathway involves rapid ligation of the two broken DNA ends, which are processed and ligated by the Ku complex and DNA–PKcs, and XRCC4–DNA ligase 4 (LIG4), respectively. Ku serves as a scaffold to recruit the core NHEJ machinery to the DNA DSB, and the Ku–DNA complex recruits DNA–PKcs, XRCC4, NHEJ1, LIG4, and APLF [112]. DNA–PKcs can be autophosphorylated or phosphorylated by ATM, while the DNA–PK complex phosphorylates H2AX, XRCC4, LIG4, and NHEJ1. The end ligation process through NHEJ requires DNA end processing [23,122,123]. The NHEJ pathway has an alternative-NHEJ (alt-NHEJ) pathway in addition to the classical-NHEJ (c-NHEJ) pathway. Compared with c-NHEJ, alt-NHEJ is a highly error-prone DNA repair pathway. The alt-NHEJ pathway exists as a backup when the c-NHEJ-repaired proteins such as DNA–PKcs and Ku70/80 are compromised, and when HR is limited. The alt-NHEJ ligates the DNA ends in the absence of the c-NHEJ factors (DNA–PKcs, XRCC4, and LIG4,). The alt-NHEJ pathway involves PARP-1, XRCC1, and DNA ligase 3(LIG3). PARP-1 is thought to bind the DNA ends and stimulate synapsis to ligation to the XRCC1/LIG3 complex [124,125].

### 7.3. DSB Repair Pathways and MM

Defects in HRR are observed in some cancers, including breast and ovarian cancers [126,127]. Hereditary *BRCA1* and *BRCA2* mutations result in increased sensitivity to DNA-damaging agents [127,128]. These observations suggest that the modulation of HRR in HR-proficient tumor cells might sensitize them to cancer chemotherapy [126]. In contrast to breast cancer, familial MM is extremely rare. However, familial MM was reported to bear a nonsense mutation in exon 27 of *BRCA2* that corresponds to a Lys 3326 Stop substitution predicted to cause the loss of the final 93 amino acids of the BRCA2 protein. This seems to be a genetic basis for at least a small fraction of MM cases [129]. Moreover, the transcript and protein levels of HR-related genes are elevated in MM cell lines and patient samples relative to normal plasma cells. These HR-related genes encode endo- and exonucleases, helicases, HsRAD51D (implicated in homologous pairing, strand exchange, and telomere maintenance), the HsRAD51 paralog XRCC3, and RAD50 [17]. Recombination activity is also elevated in myeloma cell lines and primary myeloma cells, being 11-fold greater than that observed in normal cells, as determined by a plasmid-based assay for HR [17]. High activity of HR increases the mutation rate and progressive accumulation of genetic variation over time. In MM cells, HR inhibitors induce apoptosis, and inhibition of HR activity by small inhibitory RNA targeting recombinase leads to a significant reduction in the acquisition of new genetic changes. Moreover, they suggested that high levels of RAD51 and increased HR lead to the genomic instability characteristic of the disease [130]. The acquired melphalan-resistant myeloma cell line was found to express FANCF and RAD51C, which are involved in the FA/BRCA pathway and ICL repair [130]. A clinical database showed that a higher expression of BRCA1, DNA-PK, and PARP1 was associated with poor prognosis of patients with MM treated with HDM and ASCT [131]. Furthermore, SNPs within *PARP*, *RAD51*, *MUTYH*, *OGG1*, *PCNA*, *TPMT*, and *XPC* were associated with disease progression in patients with MM who underwent HDM treatment and ASCT [44].

A large number of murine studies have demonstrated the critical role of NHEJ core proteins as well as HR mediators in maintaining genomic integrity. NHEJ core protein/*TP53* double knockout mice develop lymphomas with *IGH*/*MYC* translocations and aneuploidy [132]. NHEJ activity was found to be corrupted in RPMI-8226 cells and functional in U266 and OPM2 MM cell lines [133]. However, the association between NHEJ defects and radiation sensitivity is unclear. A gene polymorphism study on relapsed/refractory MM treated with single-agent thalidomide showed that the response rate was higher in patients with SNPs in *ERCC1*, *ERCC5*, or *XRCC5* (encoding KU80). A longer OS was associated with SNPs in *ERCC1* and *XRCC5* [134]. In addition, a correlation between polymorphisms in or aberrant expression of the *XRCC4*, *XRCC6* (encoding KU70), *DCLRE1C*, and *LIG4* genes and risk of MM development has also been reported [135,136]. High expression of the *XRCC5* and *DCLRE1C* genes was associated with poor prognosis in patients with MM [137], and upregulation of DCLRE1C, DNA–PKcs, and XRCC4 proteins has been reported in the patients’ MM cells [137]. NSD2 is the key molecular target in t(4;14) myeloma and is endowed with histone methyltransferase activity. Interestingly, NSD2 loss results in decreased expression of DNA repair genes, including *RAD51*, *TP53BP1*, and *XRCC4*, and enhanced γ-H2AX. Conversely, NSD2 overexpression is associated with increased DNA repair efficiency [138]. Given the association of t(4;14) with poor outcome in MM, this dysregulated DNA repair activity may play a role in drug resistance.

Alt-NHEJ is crucial for genomic instability and survival of MM cells. LIG3 mRNA expression in MM patients correlates with shorter survival and even increases with more advanced stage of disease. Higher LIG3 protein levels were detected in AMO1 bortezomib-resistant cells as compared to the AMO1 bortezomib-sensitive cells. LIG3 knockdown strongly increased DNA damage and finally inhibits MM cell growth in vitro and in vivo. Further, expression of miR-22 impairs growth and survival of MM cells, via targeting LIG3 [125].

## 8. FA Pathway

The FA pathway is a DNA damage-activated pathway required for the repair of ICLs. FA is a recessively inherited disorder characterized by bone marrow failure, developmental abnormalities, inability to repair ICLs, and a high incidence of malignancies [139]. ICLs are chemical bridges between two strands of DNA, leading to DNA replication and translation blocking [140]. Currently, there are 22 verified FA genes, with a possible additional gene, *RAD51C*, that make up the FA pathway. This pathway orchestrates the detection and removal of ICLs through the combined actions of NER and HR, with minor contributions of other DNA repair pathways [139,141]. Melphalan-resistant myeloma cell lines consistently express FANCF and RAD51C, which are involved in the FA/BRCA pathway for ICL repair [130], and depletion of FANCF overcomes melphalan resistance [130]. NF-κB transcriptionally regulates the FA/BRCA pathway. NF–κB, induced by melphalan, is constitutively activated in MM. In addition, the NF–κB pathway is frequently dysregulated in MM and plays a central role in the survival, proliferation, and drug resistance of MM cells.

Gene expression profile analyses of patients treated with HDM and ASCT revealed the prognostic value of genes involved in the NHEJ, HR, FA, NER, MMR, and BER pathways (Table 1). Seventeen and five out of 84 genes displayed bad and good prognostic value, respectively, for both event-free and overall survival, according to the data of patients included in this cohort. These 22 prognostic genes included: five encoding NHEJ proteins (three with bad prognostic value: *NSD2*, *RIF1*, and *XRCC5* (KU80), and two with good prognostic value: *PNKP* and *POLL*); six encoding HR proteins (five with bad prognostic value: *EXO1*, *BLM*, *RPA3*, *RAD51*, and *MRE11*, and one with good prognostic value: *ATM*); three encoding FA proteins with bad prognostic value (*RMI1*, *FANCI*, and *FANCA*); eight encoding NER proteins (six with bad prognostic value: *PCNA*, *RPA3*, *LIG3*, *POLD3*, *ERCC4*, and *POLD1*, and two with good prognostic value: *ERCC1* and *ERCC5*); two encoding MMR proteins with bad prognostic value (*EXO1* and *MSH2*); and one encoding a BER protein with bad prognostic value (*LIG3*). These findings demonstrate that these DNA repair pathways are important mechanisms in myeloma cells for melphalan treatment [142].

## 9. Epigenetic Machinery and DNA Damage

Cancer cells exhibit a high level of epigenetic modifications, which modulate expression of genes, as well as cellular processes and tumorigenesis. Epigenetic changes, such as aberrant DNA methylation and histone modification profiles, and abnormal non-coding RNA expression are considered as important in contributing to the pathogenesis of MM [143]. DNA methylation studies have shown that cancer cells are characterized by a global DNA hypomethylation and hypermethylation of tumor suppressor genes. Global hypomethylation results in the genome-wide loss of DNA methylation and subsequent genomic instability [143,144]. In MM, global DNA hypomethylation correlates with disease progression and poor prognosis [145,146]. Several studies have suggested that DNA methylation may play a role in maintaining genome stability and that DNA hypomethylation in cancer cells is related to genomic aberrations [147]. Inhibition of DNA methylation using DNA methyltransferases inhibitors (DNMTi) 5-azacytidine (AZA) and 2-deoxy-5-aza-cytidine (Decitabine) have been shown to exert anti-MM effects. *DNMT1* was reported to show an increasing median expression from the controls (101.8) to MM cells (143.3) and myeloma cell lines (1101) [145]. The cytotoxic effects of decitabine can then be explained by two modes of action [148,149]. First, trapping of DNMT enzymes leads to depletion of DNMT and the cell loses its ability to methylate DNA. The loss of methylation leads to re-activation of the silenced genes, genomic instability, and related anti-tumor effects. Second, the formation of DNA-protein adducts results in the activation of DDR that can result in apoptosis. Several studies have reported the involvement of DNA repair in response to decitabine. Chinese hamster ovary cells treated with decitabine caused DNA lesions and triggered FA-dependent HR. Decitabine induced DNA damage (gamma-H2AX foci formation) of MM cell lines, followed by a G0/G1- or G2/M-phase arrest and caspase-mediated apoptosis. Further, histone deacetylase inhibitors (HDACi), JNJ-26481585, enhanced the anti-MM effect of decitabine [145].

HDACs are involved in many biological processes, such as apoptosis, senescence, differentiation, and angiogenesis. Progression-free survival was significantly shorter in MM patients with higher levels of class I HDAC expression. Panobinostat, a pan-HDAC inhibitor, in combination with a PI and dexamethasone, has improved survival in relapsing/refractory MM patients [150]. The exact mechanism of HDACi is not completely established. Acetylation of histone induces gene expression by altering chromatin activity; acetylation of non-histone proteins affects a variety of physiological pathways, including cell growth, chromatin remodeling, and DNA replication. Further, DDR plays an important role in the biological effects of HDACis in cancer cells. Leukemia cells treated with HDACi induces DSBs, showing the appearance of phosphorylation of γH2AX nuclear foci and ATM. HDACi prevents deacetylation and disrupt the function of the DNA-repair proteins such as Ku70. HDACi acts by a transcriptional mechanism reducing DNA repair proteins such as RAD51, RAD50, DNA-PKcs, BRCA1, and BRCA2. The production of ROS is also observed after HDACi, suggesting ROS generation enhances their cytotoxic effect [151].

MicroRNAs (miRNAs) are small non-coding RNAs of 19–25 nucleotides, which regulate the gene expression by degrading or inhibiting the translation of target mRNAs, primarily via base pairing to partially or fully complementary sites in the 3′ UTR. Deregulated miRNA expression in MM patients has been associated with tumor progression, prognosis, and drug response [143,152]. Underexpression of miR-196b, miR-135b, miR-320, miR-20a, miR-19b, miR-19a, and miR-15a in MM is reported to enhance MM cell growth via overexpression of their predicted target cyclin D2 [143,153]. miRNAs can target several components of DNA damage, such as LIG3, RAD51, and BRCA1, and therefore modulate mechanisms involved in preserving genomic integrity. Long non-coding RNA (LncRNA) also participates in several biological processes, such as transcriptional gene regulation and conservation of genomic integrity. The oncogenic LncRNA NEAT1 is highly expressed in MM. NEAT1 silencing by LNA-gapmeR antisense oligonucleotide inhibits MM cell proliferation and triggers apoptosis. NEAT1 targeting silencing downregulates the HR pathway genes such as RAD51 [154]. MALAT1 is also involved in the alternative NHEJ pathway through binding with the PARP1/LIG3 complex, and regulated apoptosis via co-acting with PARP1 [155]. These findings suggest that the interplay between epigenetic machinery and the DDR has an important role in the cytotoxicity for MM by epigenetic inhibitors.

## 10. Inflammatory Microenvironment and ROS

Inflammation is a protective mechanism of the body against invading pathogens or tissue damage. However, persistent inflammation leads to a plethora of diseases, including autoimmune and metabolic disorders, atherosclerosis, cardiovascular diseases, Alzheimer’s disease, and cancer [156,157,158]. Chronic inflammation in the tumor microenvironment promotes tumor growth through cytokine production and macrophage activity [159]. The interaction between the bone marrow microenvironment and MM cells can influence cell proliferation, drug resistance, and prognosis [160,161,162,163]. Tumor-associated macrophages also play important roles in the MM microenvironment that support malignant plasma cell survival and resistance to therapy [164]. Macrophages in the tumor microenvironment release proinflammatory cytokines and contribute to genomic instability through the production of ROS and reactive nitrogen species. ROS include hydroxyl (HO*) and superoxide (O_2_*) free radicals and nonradical molecules, such as H_2_O_2_, which are less reactive than the majority of ROS [165]. These agents induce multiple types of DNA lesions, including oxidized bases, SSBs, and DSBs, which are removed by different DNA repair pathways [25]. ROS-induced DNA damage in transcriptionally active sites creates DNA/RNA hybrids, called R-loops, and requires transcription-coupled HR in a RAD52-dependent manner [166,167]. ROS are also implicated in the regulation of other DNA repair pathways, inducing other forms of DNA damage by oxidizing nucleoside bases (e.g., formation of 8-hydroxyguanine). Moreover, the DNA glycosylase OGG1 is inhibited by ROS [168]. Downstream of the sensor kinases are the transducer kinases CHK2 (activated by ATM) and CHK1 (activated by ATR), which regulate DDR, DNA repair, and cell cycle arrest. ROS activate ATM–CHK2 and ATR–CHK1 [167]. High ATR and pCHEK1 levels (ATR-CHK1) are associated with poor breast cancer survival [169]. ATR–CHK1 is also involved in drug resistance of MM, showing that ATR inhibition is strongly synergistic with melphalan, even in resistant MM cells [71]. Furthermore, DSBs can lead to ATM-mediated NF-κB activation [170]. Increased γ-H2AX levels are observed in various stages of MM development during multistep progression. γ-H2AX foci were detected in 2/5 MGUS samples, 37/40 MM samples, and 6/6 MM cell line samples during multistep progression [18]. MM cells produce high levels of ROS, as determined by the oxidation-sensitive fluorescent CellROX™ Deep Red reagent [171]. Expression of cyclin D is required for cancer cell survival and proliferation. Interestingly, almost all MM cells express one of the three cyclin D proteins, and 45% of them express cyclin D1 [172]. Cyclin D1 increases cell adhesion, migration, and cytokine secretion. This cyclin was shown to increase the production of ROS and cyclin D1-expressing MM cells, resulting in high levels of ROS without any stress. Additionally, ROS activate the ERK1/2 pathway in MM cells [172]. An increase in intracellular ROS levels may result in the activation of oncogenes and oncogenic signals, including RAS and c-MYC, which are involved in cell proliferation and inactivation of tumor suppressor genes [173]. In cancer cells, the level of ROS determines their effect. At low levels, ROS lead to reversible oxidation, thereby altering the protein activity, localization, and interactions, and control proliferation, differentiation, invasion, angiogenesis, and the drug response. At high levels, ROS induce unspecific oxidation of proteins, growth arrest, and cell death [174]. Bortezomib treatment induces ROS by decreasing intracellular glutathione in MM cells [175]. Furthermore, overproduction of ROS upon treatment with auranofin, an inhibitor of the antioxidant thioredoxin, increases the sensitivity to bortezomib [171]. Hence, ROS-mediated signaling pathways can promote tumor growth by inducing DNA damage and genomic instability and activate pro-oncogenic signaling, which causes cancer progression and survival. Due to the selective pressure induced by sustained ROS production, cancer cells have developed an efficient mechanism of ROS detoxification [176,177]. These findings suggest that inflammation promotes genomic instability synergistically with ROS production.

## 11. Hypoxia

Hypoxia is a non-physiological level of oxygen tension that is common in the majority of malignant tumors and a common feature of the BM microenvironment. A hypoxic tumor microenvironment plays a significant role in cancer progression and therapy resistance [178,179]. HIFs actively participate in the hypoxic response by regulating downstream target genes. A large number of genes and miRNAs in myeloma cells are targets of HIF1-α. For instance, miR-210 is most consistently induced under hypoxic conditions. HIF1-α directly binds to HREs on the proximal miR-210 promoter [152,153]. Furthermore, hypoxia-induced miR-210 increases the mRNA levels of *VLA4*, *CXCR4*, *IL6*, and *TGFB* in MM cells [180]. miR-210 expression is significantly correlated with that of the hypoxia-inducible genes in myeloma cells [179]. A number of chemotherapeutic drugs, including vincristine, doxorubicin, and cisplatin, have been shown to be less effective when exposed to a hypoxic environment [181]. miR-210 also increases the hypoxia-induced resistance of MM cells to melphalan [179].

As for ROS, hypoxia is known to increase its production in cells. ROS generated at mitochondrial complex III stabilize HIF1-α during hypoxia [182,183,184]. Many DDR pathways, including the HR, NHEJ, MMR, and FA pathways, have been shown to suffer alterations under hypoxia [185,186]. In particular, hypoxia alters the DSB repair pathways: DNA–PKcs, KU70/80, BRCA1, and RAD51 contribute to genomic instability and a mutator phenotype [185,186,187]. Hypoxia also contributes to altered DDR by modulating the HR and NHEJ repair proteins, leading to less sensitivity to DNA damage.

## 12. Cellular Metabolites

Cancer has its cellular changes contributing to tumor growth and cell proliferation. Both glycolysis and glutaminolysis result in energy production and nucleotide synthesis. Enhancement of glycolysis and glutaminolysis are found in MM cells compared to normal cells. Hexokinase II (HKII), which is one of four HKs isoforms, is a widely overexpressed enzyme in several cancers, including MM [188]. 3-bromopyruvate (3BrPA), an inhibitor of HKII, promptly suppresses ATP production and induces cell death in MM cells [189]. Glutamine is known as a non-essential amino acid playing a crucial role in different mechanisms in the human organism. MM cells lack glutamine synthetase and therefore depend upon the absorption of extracellular glutamine, resulting in cytotoxic effects when glutamine is depleted. Interestingly, the hypoxic environment of the BM is dependent on glutamine, suggesting glutamine transport and metabolism as therapeutic targets in MM cells [190].

EZH2 is a histone methyltransferase acting primarily at H3K27, where it catalyzes the conversion to a tri-methylated mark (H3K27me3), which is overexpressed in multiple myeloma. High EZH2 mRNA expression in MM patients is associated with poor outcomes and high-risk clinical features. This posttranslational modification by EZH2 enhances H3K27 trimethylation, an important determinant in NHEJ repair. Histone H3K27 methylation modulates the dynamics of FANCD2 on chromatin to facilitate NHEJ and genome stability [191,192].

Furthermore, recent studies have shown the interplay between cellular metabolism and the DDR in cancer. Cancer cells show an alteration of the DDR coupled with modifications in cellular metabolism. Glutamine, aspartate, and other nutrients are essential for de novo nucleotide synthesis, which dictates the availability of the nucleotide pool, and thereby influences DNA repair and replication. The increased proliferation of cancer cells can deplete the dNTP pools, causing replication stress and, consequently, DSBs. Imbalances in nucleotide pools cause misincorporations during replication, which can lead to mutations [193].

## 13. Different DDR Pathways and Their Associated Inhibitors in MM

Repair of DNA is critical for normal cellular function. However, tumor cells use DNA repair pathways to develop resistance to chemotherapy. High expression of DNA repair genes is frequently associated with poor prognosis. Therefore, inhibiting the DNA repair pathway may overcome this drug resistance and poor prognosis. Many inhibitors targeting the DNA repair pathways (APEX1, PARP1, NER, HR, c-NHEJ, alt-NHEJ, ATR, MGMT, and HDAC) and cell-cycle checkpoints (CHEK1 and CHEK2) have now been developed and might be useful to induce MM cells apoptosis in combination with DNA damage-inducing drugs (Table 2).

## 14. Conclusions

The tumor microenvironment and abnormal DNA repair function affect genetic instability in MM. Moreover, tumor microenvironment factors, such as hypoxia and inflammation, as well as chemotherapeutic agents, profoundly influence the DNA repair pathways. MM cells with dysregulated DNA repair pathways are thus dependent on the remainder functional DNA repair mechanisms. As demonstrated in this review, the relationship between tumor microenvironment and the DDR is complex; however, novel strategies in MM may be possible based on changes in DDR in MM.

## Figures and Tables

**Figure 1 cancers-13-00504-f001:**
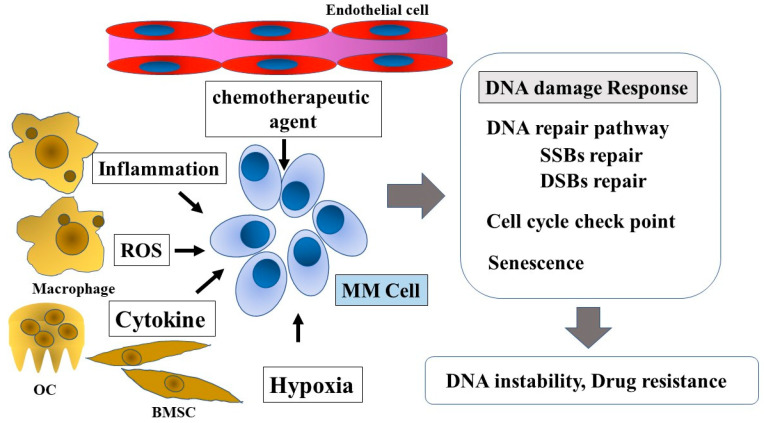
DNA damage response in multiple myeloma and the tumor microenvironment. Tumor microenvironment factors, such as chemotherapeutic agents, inflammation, ROS, cytokines, genotoxic stress, and hypoxia, influence the DNA damage response in MM cells. The DNA damage response affects disease progression and drug resistance. ROS, reactive oxygen species; OC, osteoclast; BMSC, bone marrow stromal cell; MM Cell, multiple myeloma cell; SSBs, single-strand breaks; DSBs, double-strand breaks.

**Figure 2 cancers-13-00504-f002:**
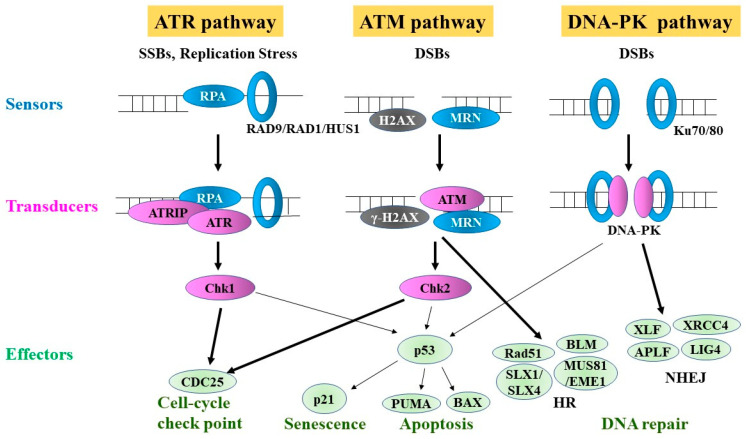
ATR, ATM and DNA-PK pathways. DNA damages are primarily detected by sensors (RPA, RAD9/RAD1/HUS1 complex, MRN complex, and Ku70/80 complex). Sensors recruit and activate transducers (ATR, ATM, DNA-PK, Chk1, and Chk2). Then, transducers phosphorylate numerous effector proteins to exert their functions, leading to senescence, cell-cycle checkpoints, apoptosis, and DNA repair. ATM and DNA-PK are responsible for the regulation of the HR and NHEJ, respectively. SSBs, single-strand breaks; DSBs, double-strand breaks; HR, homologous recombination; NHEJ, non-homologous end joining.

**Table 1 cancers-13-00504-t001:** DNA repair genes whose expression is assiciated with prognostic value. Of 84 genes, 22 had prognostic value for both overall surivival (OS) and event free survival (EFS). NHEJ, non-homologous end joining; HRR, homologous recombination repair MMR, mismatch repair, FA, Fanconi anemia pathway; NER, Nucleotide excision repair; BER, Base excision repair; HR, hazard ratio.

Pathway	Gene Name	Prognostic Value	HR (OS)	HR (EFS)
NHEJ	*NSD2*	BAD	3.7	2.8
NHEJ	*RIF1*	BAD	3.2	2.3
NHEJ	*XRCC5*	BAD	2.9	2.5
NHEJ	*PNKP*	GOOD	0.4	0.5
NHEJ	*POLL*	GOOD	0.3	0.5
HRR/MMR	*EXO1*	BAD	3.9	1.8
HRR/MMR	*BLM*	BAD	2.9	1.8
HRR/NER	*RPA3*	BAD	3.2	3.1
HRR	*RAD51*	BAD	2.8	1.7
HRR	*MRE11*	BAD	2.1	1.8
HRR	*ATM*	GOOD	0.5	0.6
FA	*RMI1*	BAD	5	3
FA	*FANCI*	BAD	3.5	2.4
FA	*FANCA*	BAD	2.2	2.4
NER	*PCNA*	BAD	4.5	2.2
NER/HRR	*RPA3*	BAD	3.2	3.1
NER/BER/a-NHEJ	*LIG3*	BAD	2.6	2
NER	*POLD3*	BAD	6.3	2.1
NER	*ERCC4*	BAD	2.5	2.2
NER	*POLD1*	BAD	2.4	2
NER	*ERCC1*	GOOD	0.4	0.4
NER	*ERCC5*	GOOD	0.5	0.5
MMR/HRR	*EXO1*	BAD	3.9	1.8
MMR	*MSH2*	BAD	2.7	1.6
BER/NER/a-NHEJ	*LIG3*	BAD	2.6	2

**Table 2 cancers-13-00504-t002:** Different DDR pathways and their associated inhibitors in MM.

Genes	DDR Pathway	Expression	OS	Drug Resistance	Inhibitors		References
*APE1*	BER (HR)	Increased	Poor	Yes	API3	Sensitize Mel	PMID: 28938675
*APE2*	BER	Increased	Poor				PMID: 28938675
*PARP1*	BER, Alt-NHEJ	Increased	Poor	Yes	Olaparib, PJ34	Sensitize Mel	PMID: 24928009, PMID: 32079692
*ERCC3*	NER	Increased	Poor	Yes	Spironolactone, Triptolide	Sensitize Mel	PMID: 28588253
*RAD51*	HR	Increased	Poor	Yes	B02	Sensitize Mel	PMID: 25996477
*DCLRE1C*	c-NHEJ	Increased	Poor				PMID: 23966156
*XRCC5*	c-NHEJ	Increased	Poor				PMID: 23966156
*LIG3*	BER, NER, alt-NHEJ	Increased	Poor	Yes	miR-22	Sensitize Bor	PMID: 30120376
*ATR*	Signaling				VX-970	Sensitize Mel	PMID: 33054085
*CHEK2*	Effector				Dinaciclib	Sensitize PARP	PMID: 26719576
*DNMT1*	DNA methyltransferase	Increased			Decitabine	Sensitize HR	PMID: 24833108
*HDAC1*	Histone deacetylase	Increased	Poor	Yes	Panobinostat	Sensitize Bor	PMID: 32267687

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
