# Peer review of "DNA Damage Response in Multiple Myeloma: The Role of the Tumor Microenvironment"

_cancers, 2021, doi:10.3390/cancers13030504_

Round 1

Reviewer 1 Report

This is a very good review on DNA damage in myeloma. The authors provide an excellent and extensive description of all DNA damage response pathway with an overlook of the major findings in the context of myeloma. The manuscript is suitable for publication in its actual format with only few minor comments/corrections.

Comments:

- Lesions affecting the epigenetic machinery are important features of myeloma and due to their functional nature, epigenetic enzyme may play an important role in myeloma pathogenesis and drug resistance. For example, epigenetic drugs in clinical trial and approved for MM, such as Decitabin and HDAC inhibitors, has been shown to target MM by inducing DNA damage. Some literature about the subject would be appropriate, perhaps in the DSB repair pathway section.

- Line 386, typo “IGH/MIC”

- Line 291, Olaparid is an PARP1 inhibitor, not inducer. The phrase line 292-294 is no properly written.

- It would be good to add that NSD2 (MMSET), one of the most important protein found to be affected in MM, is a target of PARP1 and NSD2 PARylation affects DNA repair.

- All gene name should respect the HUGO Gene Nomenclature Committee (HGNC). For example,“MMSET” is from the old nomenclature, the gene should be name NSD2.

Author Response

This is a very good review on DNA damage in myeloma. The authors provide an excellent and extensive description of all DNA damage response pathway with an overlook of the major findings in the context of myeloma. The manuscript is suitable for publication in its actual format with only few minor comments/corrections.

Comments:

- Lesions affecting the epigenetic machinery are important features of myeloma and due to their functional nature, epigenetic enzyme may play an important role in myeloma pathogenesis and drug resistance. For example, epigenetic drugs in clinical trial and approved for MM, such as Decitabin and HDAC inhibitors, has been shown to target MM by inducing DNA damage. Some literature about the subject would be appropriate, perhaps in the DSB repair pathway section.

We would appreciate your thoughtful comments. As you kindly suggested, the manuscript was revised to add the description of epigenetic machinery in MM. We added the section #9 Epigenetic machinery and DNA damage to the manuscript. L.472-L.530

- Line 386, typo “IGH/MIC”

Thank you for your kind comment. Changed IGH/MIC to IGH/MYC.

- Line 291, Olaparid is an PARP1 inhibitor, not inducer. The phrase line 292-294 is no properly written.

Thank you for your kind comment. Changed PARP inducer to PARP inhibitor. I revised the sentences according to your comments. “Moreover, PARP1 knockdown or olaparib was shown to result in significant inhibition of tumor growth using xenografts of human MM cells.” L.313-L.315

- It would be good to add that NSD2 (MMSET), one of the most important protein found to be affected in MM, is a target of PARP1 and NSD2 PARylation affects DNA repair.

Thank you for your useful comments. We added the description of PARylation of NDS2 in BER pathway. L.299-L.303

- All gene name should respect the HUGO Gene Nomenclature Committee (HGNC). For example,“MMSET” is from the old nomenclature, the gene should be name NSD2.

Thank you for your precise comments. I revised the gene names as your kindly suggested.

MMSET→NSD2, cylinD1→CCDN1, cylinD3→CCDN3, cMAF→MAF, NBS1→NBN, RAD9→RAD9A, CHK1→CHEK1, CHK2→CHEK2, APE1→APEX1, APE2→APEX2, NF-κB→NFKB1, HIF-1α→HIF1A, XLF→NHEJ1, MYH→MUTYH, FANCO/RAD51C→RAD51C, ARTEMIS→DCLRE1C, MRE11A→MRE11

Reviewer 2 Report

This paper written by Takayuki Saitoh  and Tsukasa Oda is a excellent review paper on DNA damage study in MM. I was very honored to read and review it. 

The authors supplied sufficient materials for the readers to understand the study and outlook in DDR in MM. It is well organized and stated in order. I think it's perfect to be published in Cancers which will inspire and guide the study on DDR in MM. 

In addition, I just found some small mistakes on language spelling, such as line 14 "genomic" which missed " e".

Author Response

This paper written by Takayuki Saitoh  and Tsukasa Oda is a excellent review paper on DNA damage study in MM. I was very honored to read and review it. 

The authors supplied sufficient materials for the readers to understand the study and outlook in DDR in MM. It is well organized and stated in order. I think it's perfect to be published in Cancers which will inspire and guide the study on DDR in MM. 

In addition, I just found some small mistakes on language spelling, such as line 14 "genomic" which missed " e".

I appreciate your kind comments. I changed gnomic to genomic in Line 14.

Reviewer 3 Report

The review is interesting and well written. It reads well. I have only few comments to some parts and strongly suggest to include a table where authors can describe specifically in MM, which changes in DDR occur and are associated e.g. with worse PFS, OS or drug resistance.

 Simple summary: line14, please correct gnomic to genomic instability.

Page 4, line 136-137. I think it is not correct to state that PIs induce DNA damage, as this implies they directly induce changes in the DNA, which is not true. Authors should reformulate it in line with the following lines where they discuss that PI affects DNA repair via depletion of free Ub pool that is critical for further protein ubiquitination for building DNA repair foci through protein recruitment and degradation

Page 4, line 158-159. The sentence : “Furthermore, DNA repair activity is required to counteract oxidative DNA damage in the tumor microenvironment.” implies that there is a lot of DNA damage in the neighboring cells due to oxidative DNA damage. But I guess he authors wanted to say that tumor microenvironment induces oxidative DNA damage in MM cells (due to hypoxia etc..). Thus please rewrite so that it is more logic, such as: Furthermore, DNA repair activity is required to counteract oxidative DNA damage in tumor cells induced by the tumor microenvironment.

Page 6, line 216-219. The authors describe here interesting findings for the compound VX-970 in MM. However, in the reference 65, which the authors cite at the end of these findings is no information about these findings.

In the conclusions section, authors should postulate which novel treatment strategies in MM may be possible, based on changes in DDR in MM.

Author Response

Comments and Suggestions for Authors

Reviewer 3

The review is interesting and well written. It reads well. I have only few comments to some parts and strongly suggest to include a table where authors can describe specifically in MM, which changes in DDR occur and are associated e.g. with worse PFS, OS or drug resistance.

Thank you for your useful comments. I added the table2 about the association between the DDR and clinical features in MM as you kindly suggested. L.638-L.662

 Simple summary: line14, please correct gnomic to genomic instability.

Thank you for your kind comments. I changed gnomic to genomic in Line 14.

Page 4, line 136-137. I think it is not correct to state that PIs induce DNA damage, as this implies they directly induce changes in the DNA, which is not true. Authors should reformulate it in line with the following lines where they discuss that PI affects DNA repair via depletion of free Ub pool that is critical for further protein ubiquitination for building DNA repair foci through protein recruitment and degradation.

Thank you for your kind comments. We revised the manuscript according to your advice. L.142

Page 4, line 158-159. The sentence : “Furthermore, DNA repair activity is required to counteract oxidative DNA damage in the tumor microenvironment.” implies that there is a lot of DNA damage in the neighboring cells due to oxidative DNA damage. But I guess he authors wanted to say that tumor microenvironment induces oxidative DNA damage in MM cells (due to hypoxia etc..). Thus please rewrite so that it is more logic, such as: Furthermore, DNA repair activity is required to counteract oxidative DNA damage in tumor cells induced by the tumor microenvironment.

Thank you for your thoughtful comments. I completely agree with you. The manuscript was revised as you suggested. L.171

Page 6, line 216-219. The authors describe here interesting findings for the compound VX-970 in MM. However, in the reference 65, which the authors cite at the end of these findings is no information about these findings.

I appreciate your precise comments. The reference number was wrong. I changed the reference number about VX-970. Botrugno OA, Bianchessi S, Zambroni D, Frenquelli M, Belloni D, Bongiovanni L, Girlanda S, Di Terlizzi S, Ferrarini M, Ferrero E, Ponzoni M, Marcatti M, Tonon G. ATR addiction in multiple myeloma: synthetic lethal approaches exploiting established therapies. Haematologica. 2019 Nov 14;105(10):2440-2447.

In the conclusions section, authors should postulate which novel treatment strategies in MM may be possible, based on changes in DDR in MM.

I appreciate your kind comments. I changed the description in the conclusion section. L.670

Reviewer 4 Report

In this paper the authors review the main DNA repair mechanisms and their regulation by the microenvironment in multiple myeloma.

The work is well written and fluent in reading. Here are some weaknesses:

I would suggest that the authors make one/two tables where they summarize the main points that they describe and discuss more in detail in the text.

Lines 58-59 - It is necessary to rephrase the sentence to make it clearer.

Lines 59-60 – The authors should discuss more on “the role of DNA repair in regulating the metabolism of MM cells”.

Lines 114-116 - Expand and clarify the concept about the role of CIN adding more recent references, such as Neuse CJ et al. 2020.

Lines 139-141 - The result reported refers to a work published 10 years ago: please review in the light of more recent data.

Lines 418-430 - Gene expression profile data described in the text are difficult to follow, it might help to transform in a table or add a table to the text.

Lines 481-482 - The reference does not match with what is described in the text. Please modify by updating with more recent references.

Author Response

In this paper the authors review the main DNA repair mechanisms and their regulation by the microenvironment in multiple myeloma. The work is well written and fluent in reading. Here are some weaknesses: I would suggest that the authors make one/two tables where they summarize the main points that they describe and discuss more in detail in the text.

We appreciate you provide thoughtful comments. We added the table2 and discuss the association between the DDR genes and clinical features of MM patients. L.638

Lines 58-59 - It is necessary to rephrase the sentence to make it clearer.

Thank you for your comments. I agree with you. I have made the sentence clear using the DDR. I change to “This review provides an overview of DDR, with a special focus on their function of MM” L.58   

Lines 59-60 – The authors should discuss more on “the role of DNA repair in regulating the metabolism of MM cells”.

We would appreciate your kind comments. As you kindly suggested, the manuscript was revised to add the description of cell metabolite and the DDR in MM. We added the section #12 Cellular metabolites to the manuscript. L.609-637

Lines 114-116 - Expand and clarify the concept about the role of CIN adding more recent references, such as Neuse CJ et al. 2020.

Thank you very much for your useful comments about the role of CIN. We added and the description of the concept of and the role of CIN by referring to the manuscript as you suggested. L.94-

Lines 139-141 - The result reported refers to a work published 10 years ago: please review in the light of more recent data.

Thank you very much for your kind comments. We changed the description of clinical study with bortezomib and high-dose melphalan vs. high-dose melphalan as conditioning regimen ASCT. We also added the invitro data of bortezomib and melphalan. L.150-154

Lines 418-430 - Gene expression profile data described in the text are difficult to follow, it might help to transform in a table or add a table to the text.

Thank you very much for your useful comments. We added table1 for the expression of DNA repair genes, as you kindly suggested. L.470

Lines 481-482 - The reference does not match with what is described in the text. Please modify by updating with more recent references.

Thank you for your kind comments. I changed the reference.

Round 2

Reviewer 4 Report

Most of my previous critiques have been addressed. However, some changes are required before publication.

References 53 is the same as reference 54. According to the text, reference 54 should refer to a more recent work, as suggested in the review process.

Lines 573-575: Move the reference 180 to the end of the sentence "Bortezomib treatment induces ROS by decreasing intracellular glutathione 573 in MM cells" and delete it from line 575 because, as already indicated in the review process, I couldn't find mention of auranofin in the ref 180.

Lines 618-620: Please correct the sentence.

Author Response

Dear Reviewer 4,

Thank you very much for your contribution of our manuscript.

References 53 is the same as reference 54. According to the text, reference 54 should refer to a more recent work, as suggested in the review process.

Thank you for your suggestions. I changed the reference#54

54   Murielle Roussel, Benjamin Hebraud, Valerie Lauwers-Cances, Margaret Macro, Xavier Leleu, Cyrille Hulin, Lionel Karlin, Bruno Royer, Aurore Perrot, Philippe Moreau, et al. Bortezomib and High-Dose Melphalan Vs. High-Dose Melphalan As Conditioning Regimen before Autologous Stem Cell Transplantation in De Novo Multiple Myeloma Patients: A Phase 3 Study of the Intergroupe Francophone Du Myelome (IFM 2014-02) Blood. 2017;130(Supplement 1): 398.

Lines 573-575: Move the reference 180 to the end of the sentence "Bortezomib treatment induces ROS by decreasing intracellular glutathione 573 in MM cells" and delete it from line 575 because, as already indicated in the review process, I couldn't find mention of auranofin in the ref 180.

Thank you for your suggestions.

1)I moved the reference180 to the sentence "Bortezomib treatment induces ROS by decreasing intracellular glutathione in MM cells".

Bortezomib treatment induces ROS by decreasing intracellular glutathione in MM cells[180].

2)I deleted the sentence "Many antioxidant enzymes, including thioredoxin (TRX1), thioredoxin reductase 1 (TRXR1), [Cu-Zn] superoxide dismutase 1 (SOD1), glutaredoxin 2/3 (GLRX2/3), and peroxiredoxin 6 (PRDX6), are overexpressed in MM cells. MM exhibits higher intrinsic oxidative stress than normal cells and is adapted to this redox status by upregulating antioxidant enzymes."

3) I deleted the reference 180 as follows in the following setence.

Furthermore, overproduction of ROS upon treatment with auranofin, an inhibitor of the antioxidant thioredoxin, increases the sensitivity to bortezomib [180, 181]

Lines 618-620: Please correct the sentence.

Thank you for your pointing it out. I erased unnecessary sentence in order to make it simple. MM cells show an excess of NH4+ produced from glutamine, which leads to the assumption that MM cells are glutamine addicted.